# Circular RNAs in Acute Kidney Injury: Roles in Pathophysiology and Implications for Clinical Management

**DOI:** 10.3390/ijms23158509

**Published:** 2022-07-31

**Authors:** Benjamin Y. F. So, Desmond Y. H. Yap, Tak Mao Chan

**Affiliations:** Division of Nephrology, Department of Medicine, Queen Mary Hospital, The University of Hong Kong, Hong Kong, China; desmondy@hku.hk (D.Y.H.Y.); dtmchan@hku.hk (T.M.C.)

**Keywords:** acute kidney injury, circular RNAs, sepsis, ischaemia-reperfusion injury, drugs

## Abstract

Acute kidney injury (AKI) is a common clinical condition, results in patient morbidity and mortality, and incurs considerable health care costs. Sepsis, ischaemia-reperfusion injury (IRI) and drug nephrotoxicity are the leading causes. Mounting evidence suggests that perturbations in circular RNAs (circRNAs) are observed in AKI of various aetiologies, and have pathogenic significance. Aberrant circRNA expressions can cause altered intracellular signalling, exaggerated oxidative stress, increased cellular apoptosis, excess inflammation, and tissue injury in AKI due to sepsis or IRI. While circRNAs are dysregulated in drug-induced AKI, their roles in pathogenesis are less well-characterised. CircRNAs also show potential for clinical application in diagnosis, prognostication, monitoring, and treatment. Prospective observational studies are needed to investigate the role of circRNAs in the clinical management of AKI, with special focus on the safety of therapeutic interventions targeting circRNAs and the avoidance of untoward off-target effects.

## 1. Introduction

Acute kidney injury (AKI) is a syndrome of rapid deterioration in kidney function within hours to days. AKI is common and occurs in up to 10–20% of all hospitalised inpatients, and is associated with significant morbidity, mortality, and health care costs [1]. Crucially, even small perturbations in kidney function among hospitalised patients have been associated with a variety of short- and long-term adverse clinical outcomes, increased length of stay, and greater utilisation of limited health care resources [2]. Short- and long-term mortality is increased drastically in patients with AKI, and recovery from AKI is often incomplete [3,4]. Survivors of AKI are at an escalated risk of developing chronic kidney disease (CKD), cardiovascular disease, and other related comorbidities [5,6].

Despite its clinical significance, there is limited progress to date in therapeutic interventions for AKI per se, and management is directed at the underlying causes. Few disease-modifying therapies are available for most causes of AKI, and therapy is largely supportive in nature. Part of the reason may be that AKI is a heterogeneous, broadly defined syndrome with various causes, each with different underlying pathophysiology. The most common causes in developed countries nowadays include sepsis, haemodynamic insults such as hypotension with ischaemia-reperfusion injury (IRI), nephrotoxic drugs, and urinary tract obstruction [1]. Multiple definitions of AKI exist in the literature. Nowadays, the most commonly used definition of AKI is the one defined by the Kidney Disease: Improving Global Outcomes (KDIGO) initiative, which is based on documented increases in serum creatinine within a prespecified time frame and/or a reduction in urine output (Figure 1) [7]. Regardless of the definition used, conceptually, AKI should be viewed as a particularly abrupt presentation of the impairment of kidney function due to exposure to an acute insult. This is contrasted with CKD, which by definition requires demonstration of renal impairment for at least 3 months [8]. While clinical management in AKI aims to stop further injury and potentially reverse the renal impairment, a diagnosis of CKD implies a significant element of irreversibility, and treatment is often focused on retarding progression and the mitigation of complications.

Certain causes of AKI, such as sepsis, remain associated with unacceptably high rates of mortality [9]. There is clearly a pressing need to investigate the common causes of AKI with the aim of designing better diagnostic and therapeutic tools. In this context, a growing body of research has focused not only on the clinical aspects of these conditions, but also on the genetic and epigenetic landscape underpinning these clinical presentations. For example, certain genetic polymorphisms have been associated with susceptibility to particular nephrotoxic insults and the development of AKI [10]. Non-coding RNAs (ncRNAs) that modulate the expression of target mRNAs have been identified in various diseases, and there is an emerging role of epigenetics in the pathogenesis of human disease and its promise in diagnostics and therapeutics. While much research has focused on long non-coding RNAs (lncRNAs) and microRNAs (miRNAs) in the field of AKI [11,12], there is also accumulating data on circular RNAs (circRNAs), which may exert epigenetic modulation upstream of their target miRNAs and lncRNAs. CircRNAs have been implicated in the pathogenesis of various diseases including cancers, neurodegenerative diseases, and autoimmune diseases, highlighting their impact on and importance in various body systems, although to date, there is less published data on AKI [13,14,15]. This review will therefore focus on the evidence regarding circRNAs in different human and murine models of AKI, discuss the possible applications for the diagnosis, prognostication, and treatment of AKI, and identify key gaps in knowledge requiring further clarification and research.

## 2. CircRNAs in Health and in Kidney Disease

CircRNAs are distinguished from other linear RNAs by their structure. CircRNAs are typically single-stranded, covalently closed molecules. Typically, pre-mRNA contains exons and introns, followed by a 7-methylguanosine cap and polyadenosine tail added to its 5′ and 3′ ends, respectively. Pre-mRNA then undergoes splicing at canonical splice sites (5′-GU and 3′-AG at introns) to become mature and translatable. CircRNAs are produced by a non-canonical splicing event called back-splicing, in which a downstream 5′ splice donor site is covalently joined to an upstream 3′ splice acceptor site through a 3′5′-phosphodiester bond, resulting in a circular structure [16].

Although circRNAs were first described in viroids and viruses, they have since been found to be ubiquitous among eukaryotes including in the human transcriptome [17]. They are highly conserved among eukaryotic organisms in both sequence and expression, making them relatively easy to model and study [18]. CircRNAs formed from exons are usually localised to the cytoplasm, whereas intronic and some exonic circRNAs are localised to the nucleus. CircRNAs can regulate gene expression by influencing the transcription, the mRNA turnover, and translation by sponging RNA-binding proteins and miRNAs. They function as miRNA sponges and upregulate the mRNAs targeted by miRNAs [19]. They may also serve as transcription regulators by binding to their target gene at their synthesis locus and cause transcriptional pausing through the formation of an RNA-DNA hybrid R-loop structure [20]. CircRNAs may interact with various proteins by serving as protein sponges or decoys, or by forming scaffolds to facilitate protein complex formation and reaction [21]. Like other ncRNAs including miRNAs, circRNAs have been found in exosomes that are secreted into the blood, urine, or other body fluids [22]. CircRNAs have half-lives that are at least 2.5 times longer than linear RNAs, since their circular structure renders them resistant to exonucleases. For example, circRNAs of human mammary cells have a half-life of 18.8–23.7 h compared to only 4.0–7.4 h for their linear counterparts [23]. CircRNAs may hence be potentially more stable biomarkers of various diseases than other ncRNAs.

CircRNAs can be assayed using a variety of methods, with RNA-Seq of rRNA-depleted total RNA being the method of choice for discovering new circRNAs as it also allows for the characterisation of their linear counterparts [24]. Advances in high throughput sequencing and novel bioinformatics tools and databases have allowed for the development of genome-wide circRNA profiling, which was first described in 2012 [25]. Accordingly, circRNA analysis has also been extensively applied to the study of kidney disease. An RNA-Seq study showed that 1664 circRNAs are significantly expressed in human kidney tissues including 474 unique to the kidney [26]. Aberrant expressions of circRNAs have been reported in a variety of acute and chronic kidney disorders including diabetic kidney disease and glomerular disorders such as focal segmental glomerulosclerosis [27,28]. A genome-wide circRNA expression analysis from intensive care unit (ICU) patients showed that circulating hsa_circ_0003266 (also referred to as circRNA sponge of miR-126) was significantly upregulated in the serum of AKI patients of various causes at the inception of renal replacement therapy compared to the non-AKI controls, and was a predictor of mortality at 28 days [29]. This is consistent with previous findings that miR-126, the target of ciRs-126, mediates vasculogenic progenitor cell mobilisation via stromal cell-derived factor 1/CXCR4 pathways to promote vascular and renal recovery from IRI [30]. Clearly, circRNAs may be differentially expressed in different forms of kidney disease, and may play pathogenic, protective, or prognostic roles. Although causes of AKI were traditionally divided into pre-renal, renal, and post-renal causes, many aetiologies are pathophysiologically complex and encompass components of all three types of renal injury. Thus, the most common AKI syndromes of sepsis, IRI, drug nephrotoxicity, and urinary tract obstruction will each be explored in the following sections. Some examples of differentially expressed circRNAs in each phenotype are shown in Table 1.

## 3. CircRNA Expression in AKI Due to Sepsis

Sepsis is the most common cause of AKI in developed countries, and is associated with an exaggerated increase in the risk of mortality, especially in critically ill populations [9]. Septic AKI may occur in infection by bacteria, viruses, fungi, and parasites, and in the absence of direct invasion of the renal tissue by pathogens. Conventionally, septic AKI was thought to be due to a reduction in the global renal blood flow with tubular epithelial cell death, also known as acute tubular necrosis, and much research in this regard has therefore focused on ischaemic injury [60]. However, AKI may occur in septic patients even in the absence of hypotension. Studies have shown that blood flow is not necessarily decreased in septic AKI, and may sometimes in fact be increased [61]. Postmortem histological findings or experimental kidney biopsies showed that kidney histopathology did not appear to be as severe as expected based on the severity of renal impairment [62]. While hypoperfusion and IRI may play a limited role in the pathophysiology of septic AKI, the current understanding focuses on inflammation, microcirculatory dysfunction, and metabolic reprogramming as key pathways in septic AKI. Most of the contemporary evidence on the role of circRNAs in septic AKI focuses exclusively on inflammatory pathways, as illustrated in Figure 2.

During sepsis, inflammatory mediators such as pathogen-associated molecular patterns (PAMPs) and damage-associated molecular patterns (DAMPs) are released into the circulation. Pattern recognition receptors such as Toll-like receptors (TLRs) initiate signal transduction that eventually results in the release of proinflammatory molecules such as IL-6 and TNFα. When PAMPs and DAMPs are filtered through the glomeruli, similar receptors also take up these signals and result in an increase in oxidative stress, expression of reactive oxygen species (ROS), cellular injury and apoptosis, and mitochondrial injury [63]. Even in the absence of overt hypoperfusion related to low blood pressure, microcirculatory dysfunction may occur in septic AKI. A reduction in capillary density is observed in septic patients with AKI, and there is often a reduction in capillaries with continuous flow and an increase in capillaries with intermittent or stop flow [64]. Such microcirculatory dysfunction may be related to endothelial injury from inflammatory mediators, autonomic dysfunction, and activation of the coagulation cascade as well as other mechanisms [65,66]. Changes in the haemodynamics of the microcirculation have been studied in septic kidneys. Intrarenal shunting, with blood bypassing glomerular filtration and directly passing from the afferent to efferent arteriole, may occur in sepsis, although the exact pathophysiology is not well-characterised [64]. Blood may also be preferentially redistributed away from the renal medulla during sepsis, resulting in segmental ischaemia [67]. Finally, metabolic reprogramming is evident in septic AKI, as cells adapt patterns of energy utilisation to prioritise cellular survival over cellular function. Non-vital cellular processes that require large amounts of adenosine triphosphate (ATP) such as protein synthesis or ion transport, are deprioritised in favour of essential functions such as function of the membrane Na^+^/K^+^ pump, in order to conserve energy [68].

Two key experimental models have been created to study septic AKI, and have proven popular for the study of the epigenetic signature of AKI. Since septic AKI has a complex, multifaceted pathophysiology, these models are necessarily limited in their generalisability to the human syndrome of septic AKI, although they illustrate key features of AKI in sepsis that would require further in vivo studies. The first model of lipopolysaccharide (LPS)-induced inflammation is an in vitro model based on the use of endotoxin expressed on Gram-negative bacteria. Kidney cell lines such as Human Kidney 2 (HK-2) cells derived from human proximal tubular cells, or Normal Rat Kidney Clone 52E (NRK-52E) cells derived from rat kidney epithelial cells, exposed to LPS reliably generate large quantities of proinflammatory cytokines such as IL-1 and TNFα. However, studies have shown that the concentration of cytokines generated by the LPS-induced model is several-fold higher than those typically observed in human sepsis [69,70]. Additionally, a compensatory response typically occurs in later stages of human sepsis where the overwhelming immune response is modulated and attenuated to limit immune-mediated damage, which is not captured by the LPS-induced model [71]. Nonetheless, this model serves as an adequate simulation of the early, acute, and hyperinflammatory phase of sepsis and septic AKI. The second model is an animal model that involves the use of caecal ligation puncture (CLP) surgery on rats to induce polymicrobial sepsis. Although the cytokine profile of such a model is more closely assigned with that observed in human sepsis, kidney or lung injury is not consistently reproduced by CLP surgery [69,70]. Therefore, documentation of an increase in serum creatinine, urea, or other indices of kidney function, is necessary for studies of septic AKI using the CLP model.

Separate studies showed that circ-FANCA (hsa_circ_0040994), circ-BNIP3L (hsa_circ_0002131), circ-HIPK3, hsa_circ_0114427, and hsa_circ_0114428 were each increased in the sera of patients with septic AKI [31,32,34,35,36]. In human cell line models of septic AKI, HK-2 cells treated with LPS showed elevated cytoplasmic expression of circ-FANCA, circ-BNIP3L, hsa_circ_0114427, hsa_circ_0114428, circ-HIPK3, and circ-TLK1, whereas knockout of circ-FANCA, circ-BNIP3L, hsa_circ_0114427, hsa_circ_0114428, circ-HIPK3, and circ-TLK1, respectively, resulted in the attenuation of LPS-induced cell injury, implying a direct pathogenic role of these circRNAs [31,32,34,35,36,37]. These effects seem to be mediated via interaction with other epigenetic regulators. Circ-FANCA sponges miR-93-5p, a miRNA that is typically downregulated in models of septic AKI, leading to cellular apoptosis, inflammation, and oxidative stress [31]. Meanwhile, circ-BNIP3L targets miR-370-3p. Decreased expression of miR-370-3p in septic AKI was associated with elevated expression of MYD88, which is a downstream adaptor molecule of the TLR and IL-1 receptor families and modulates the activity of mitogen-activated protein kinase (MAPK) and NF-κB pathways [32]. Direct interaction was observed between both hsa_circ_0114427 and hsa_circ_0114428 with miR-495-3p, which regulates TRAF6 expression via the NF-κB signalling pathway [35,36]. Circ-HIPK3 sponges miR-338 and modulates the forkhead box AI axis to aggravate LPS-induced cell injury [34]. Circ-TLK1 sponges miR-106-5p and increases the action of its gene target high-mobility group box 1 (HMGB1), an inflammatory mediator secreted by a variety of immune cells [37]. Furthermore, the overexpression of circ-FANCA was also detected in exosomes isolated from LPS-treated HK-2 cells, suggesting that exosomal circ-FANCA may possibly serve as a biomarker for septic AKI as well as a potential therapeutic target [31].

In murine models, circ-RASGEF1B was upregulated in LPS-treated TCMK-1 cells and contributed to LPS-induced cellular apoptosis and inflammation by binding to and sponging miR-146a-5p. Downstream, the expression of mRNA of the miR-146a-5p target Pdk1 was elevated, in proportion to the increase in circ-RASGEF1B [38]. Like in human HK-2 cells, in a rat model of septic AKI induced by CLP, circ-TLK1 was increased compared to the control rats; knockdown of circ-TLK1 in vivo reduced urinary levels of tubular injury markers such as neutrophil gelatinase-associated lipocalin (NGAL) and kidney injury molecule-1 (KIM-1), repressed oxidative stress markers such as ROS and proinflammatory cytokines such as IL-6 and IL-1β, and improved renal function with lower serum levels of creatinine and urea [37]. Similar findings were noted for circ-HIPK3 in a murine model of AKI induced by *Candida albicans*, where the downregulation of circ-HIPK3 ameliorated the inflammatory response to sepsis, with reduced levels of IL-6 and TNF-α but increased levels of IL-10. This was likely due to the interaction of circ-HIPK3 with miR-124-3p and miR-148b-3p [33].

Certain circRNAs may be downregulated in septic AKI. For example, both hsa_circ_0068,888 and hsa_circ_0091702 were reduced in the plasma of patients with septic AKI and likewise downregulated in LPS-stimulated HK-2 cells. In vitro studies showed that hsa_circ_0068,888 could sponge miR-21-5p to inhibit the expression of proinflammatory cytokines such as IL-6, TNF-α, and IL-1β, thereby protecting against LPS-induced cell injury via the NF-κB pathway [39,40]. Similarly, hsa_circ_0091702 sponges miR-545-3p, attenuates inflammatory responses, and increases THBS2, a thrombospondin responsible for mediating intercellular interactions and regulating tumour growth [40]. Likewise, circ-PRKCI was decreased in patients with septic AKI and in LPS-simulated HK-2 cells. Circ-PRKC1 likely acts as a sponge for miR-106b-5p to regulate the expression of growth factor receptor binding 2-associated binding protein 1 (GAB1), a protein that has been associated with tissue damage in acute lung injury; the overexpression of circ-PRKCI had salutary effects on cell apoptosis and inflammation [41]. In a rat CLP model of septic AKI, circ-Ttc3 was downregulated in kidney samples with the resultant overexpression of miR-148a. When circ-Ttc3 was overexpressed in the rat model, the inflammatory response was significantly attenuated, with a decrease in the serum levels of creatinine and urea, proinflammatory cytokines such as IL-6 and TNF-α, and the tubular injury marker neutrophil gelatinase-associated lipocalin (NGAL) [42]. Rat mesangial cells treated with LPS showed decreased expression of circ-MTO1 (hsa_circ_0007847); overexpression of circ-MTO1 led to sponging of miR-337 and regulation of Kruppel-like factor 6 (KLF6), a key transcription factor associated with inflammatory pathways [43]. Similar to the above examples, circ-VMA21 was reduced in patients with septic AKI and also in the LPS-stimulated HK-2 cells; upregulation of circ-VMA21 in a rat CLP model alleviated oxidation stress, cellular apoptosis, inflammation, and tissue injury in septic AKI, purportedly by functioning as a sponge of miR-9-3p [44]. Whether the upregulation of such circRNAs could be harnessed for the treatment of septic AKI remains to be investigated.

## 4. CircRNA Expression in AKI Due to Ischaemia-Reperfusion Injury

IRI to any organ is caused by a sudden, significant but temporary impairment in blood flow. In human patients, this can occur secondary to hypoperfusion of any reason; for example, this may occur with haemorrhagic shock due to torrential gastrointestinal bleeding, with hypovolaemia due to severe diarrhoea, or with reduced end-organ perfusion due to heart failure, or in the transplanted kidney [72]. As described above, septic AKI was historically thought to be due to IRI. Although these two syndromes do in fact share common features and pathophysiologic factors, AKI due to IRI carries a different molecular and genetic signature than septic AKI.

Differential hypoxia is commonly observed in IRI. The outer medulla is supplied by the vasa recta, which are relatively oxygen-depleted after supplying the energy-dependent counter-current exchange system in the loop of Henle, and these areas are therefore particularly susceptible to hypoxic/ischaemic insults, even in the absence of global hypoperfusion [73]. Drugs that impede normal renal autoregulation or tubuloglomerular feedback also contribute [72]. Then, in response to tissue ischaemia, inflammatory mediators such as IL-6 and TNFα recruit leukocytes into ischaemic tissues, triggering the further release of cytokines, ROS, proteases, and other mediators that result in tissue damage. The release of oxygen free radicals is exacerbated by reperfusion, leading to lipid peroxidation [74]. Tissue ischaemia also causes mitochondrial dysfunction due to the failure of oxidative phosphorylation. Intracellular processes dependent on ATP fail, contributing to cellular apoptosis. Mitochondrial damage is further worsened by the high concentration of ROS as well as the increase in cytosolic calcium due to failure of the Na^+^/Ca^2+^ antiporter with ATP depletion, in the post-ischaemic cell [75]. Activation of the renin–angiotensin–aldosterone system secondary to tissue ischaemia results in renal vasoconstriction, leading to further oxidative stress and the potentiation of cellular apoptosis [76]. The complement system is also activated in IRI, leading to the release of active complement end products that are proinflammatory and also stimulate the upregulation of cell adhesion molecules [77].

Renal IRI can be reliably modelled in animals. Bilateral or unilateral renal IRI murine models are established with the clamping of renal pedicle(s) for desired period(s) of ischaemia. In the unilateral renal IRI model, the contralateral kidney may be removed to reduce variability and to provide control tissue, or kept in situ to provide control tissue in studies looking at renal fibrosis from IRI and progression to CKD. The bilateral model may more closely simulate IRI in humans as hypoperfusion is typically global and bilateral [78].

As described above, certain circRNAs are upregulated in diverse populations with AKI. The exact pathophysiological roles of such circRNA signatures have been discerned by in vitro studies on human and murine cell lines. For example, hsa_circ_0023404 was greatly increased in AKI patients, and was also significantly upregulated in HK-2 cells exposed to hypoxia then reperfusion. Bioinformatic analysis suggests that hsa_circ_0023404 sponges miR-136. Agomirs of miR-136 have been shown to repress the IL-6 receptor mRNA and protein and its downstream proinflammatory effects [45]. In a murine model of renal IRI, miR-144-5p was protective against injury progression via the inhibition of cell apoptosis and the repression of Wnt/β-catenin signals; antagomirs to miR-144-5p promoted tissue apoptosis and damage. Circ-AKT3 was found to be a potential sponge to miR-144-5p that could aggravate renal IRI [46]. The microRNA miR-328-3p is downregulated in AKI with a consequent increase in the expression of Pim-1 proto-oncogene, which impacts cellular apoptosis and transcriptional activation; screening of potential circRNA sponges with bioinformatic analysis identified circ-ITGB1 (hsa_circ_0018148), which is increased in AKI, as the most likely candidate circRNA sponge [47]. Similarly, hsa_circ_001839 promoted inflammation as evidenced by TNF-α, IFN-γ, and IL-6 levels by interacting with miR-432-3p and its regulatory product nucleotide-binding oligomerization domain-like receptor containing pyrin domain 3 (NLRP3) [48]. Likewise, mmu_circ_0000943 was upregulated in mouse kidney proximal tubule cells exposed to IRI. Sponging of miR-377-3p by mmu_circ_0000943 resulted in the overexpression of early growth response 2 (Egr2) and was associated with increased cellular apoptosis, inflammation, and oxidative stress [49]. NRK-52E rat kidney epithelial cells exposed to hypoxia and reperfusion showed increased expression of circ-SNRK, which was shown to participate in the activation of the MAPK signalling pathway and promoting cell apoptosis and secretion of TNF-α and IL-6. Knockdown of circ-SNRK resulted in the reduction in renal inflammation and improvement in the serum creatinine and urea levels [50]. Conversely, circ-YAP1 was downregulated in AKI patients and in HK-2 cells subjected to IRI conditions. Overexpression of circ-YAP1 moderated IRI in vitro by acting as a sponge for miR-21-5p and activating signal transduction via the PI3K/Akt/mTOR pathway, a crucial pathway for regulating inflammatory and fibrotic responses in response to tissue injury [53].

The list of circRNAs that may be differentially expressed in renal IRI is clearly not exhaustive. One study profiled all circRNAs expressed in rat kidneys exposed to IRI compared to the controls, and showed significant differential expression of at least 26 circRNAs, associated with 26 miRNAs and their downstream genes. For example, circ-DNMT3A, circ-PLEKHA7, and circ-ME1 were decreased in rats with ischaemic AKI. Many of these miRNAs and their associated genes are associated with the PI3K/Akt/mTOR pathway, underscoring the importance of this system in modulating the response to renal IRI. Importantly, in the rat model of IRI, pre-treatment with the angiotensin-II receptor blocker losartan attenuated kidney damage, and this was associated with a reversal of the differential circRNA expression profile in the kidneys of untreated controls [54]. In another study of unilateral IRI murine kidney model 4983 circRNAs were found to be differentially expressed, and bioinformatic analysis showed that the parental genes of these circRNAs were involved in focal adhesion, adhesion junctions, and the regulation of actin cytoskeleton pathways. Two hub genes, namely, circ-SLC8A1 (mmu_circ_0000823) and circ-APOE (mmu_circ_0014064), were associated with a large number of differentially expressed miRNAs and mRNAs on bioinformatic analysis [51]. These circRNAs are also aberrantly expressed in other conditions such as solid organ tumours and CKD, where they are associated with cell proliferation, angiogenesis, cell migration, and epithelial-to-mesenchymal transition including in the induction and propagation of renal fibrosis. Therefore, it is plausible that these circRNAs are crucial not only in the pathogenesis of AKI, but also in the accrual of chronic damage and transition from AKI to CKD. These findings further highlight the importance of examining non-kidney off-target effects when investigating the therapeutic potential of agents that target circRNAs.

## 5. CircRNA Expression in Drug-Induced AKI

The pathogenesis of AKI resulting from exposure to nephrotoxic drugs depends largely on the nature of the drug in question. Though there is extensive human data regarding the putative pathophysiology of different nephrotoxins, the best characterised animal models studied in vitro include models of nephrotoxicity due to the chemotherapeutic agent cisplatin, the aminoglycoside antibiotic gentamicin, radiocontrast media, and non-steroidal anti-inflammatory drugs (NSAIDs) such as diclofenac [78]. The finer details of all of these models are beyond the scope of this review, but the models of cisplatin-induced nephrotoxicity and contrast-associated AKI will be elaborated, as epigenetic modifications including the role of circRNAs have been investigated in these two models.

Cisplatin is a platinum-based chemotherapeutic agent commonly used for various solid organ tumours in human subjects including cancers of the lung, head and neck, and testes. Cisplatin use is limited by dose-dependent nephrotoxicity, although genetics may also determine individual susceptibility to these toxicities. The mechanisms underlying cisplatin-induced AKI are complex [79]. Cisplatin is rapidly taken up into proximal tubular epithelial cells, where they accumulate and trigger inflammatory cascades such as the NF-κB and MAPK pathways and promote the generation of ROS [80,81]. High concentrations of cisplatin result in necrosis, whereas lower concentrations induce apoptosis via the external pathway mediated by death receptors, the internal pathway mediated by the mitochondria, or by endoplasmic reticulum-stress pathways [79,81]. Additionally, cisplatin potentiates renal vasoconstriction, further leading to tissue hypoxia and injury [82]. The nephrotoxic effects of cisplatin can be replicated not just in humans but also in other mammalian models including mice after systemic administration by the intravenous or intraperitoneal routes [79].

Iodinated contrast media, used widely for radiological studies in human patients, has been associated with acute impairment of kidney function, particularly in patients with pre-existing CKD. Although the syndrome is common, comparatively little is actually known about the underlying pathophysiology of contrast-associated AKI. Studies suggest that modifications in renal haemodynamics may play a key role. Iodinated contrast media appears to preferentially cause renal medullary vasoconstriction, decreasing oxygen tension in the renal medulla, the part of the nephron most susceptible to hypoxic/ischaemic insults [83,84]. How exactly contrast media leads to increased expression of vasoconstrictors such as endothelin and angiotensin II, and the reduction in vasodilators such as prostacyclin and nitric oxide is not well-understood. Additionally, the high osmotic load of iodinated contrast induces an osmotic diuresis that results in increased sodium delivery to the thick ascending loop of Henle, which dips through the renal medulla, and thereby increases oxygen demand [83,85]. Finally, contrast media may also have direct cytopathic effects on tubular and mesangial cells, although the exact mechanisms leading to cellular apoptosis are not well-characterised [86]. Increased production of ROS may play a role [52]. Animal models of contrast-associated AKI are typically established by pre-treatment with a period of dehydration followed by systemic contrast exposure [87].

Differential expression of circRNAs has been reported for cisplatin-induced AKI. The circRNA hsa_circ_0114427 was upregulated in different AKI models including cisplatin-induced AKI, and likely functions by sponging miR-494 to regulate ATF3 expression, which is associated with the downstream expression of IL-6 [55]. In a murine model induced by acute exposure to cisplatin, 368 circRNAs were differentially expressed after cisplatin treatment including 224 upregulated circRNAs and 144 downregulated ones. These circRNAs were mainly generated from exons, and were found on all murine chromosomes except chromosomes X and Y [88]. While most of these circRNAs should also function by sponging miRNAs, the exact mechanisms of most circRNAs were not well-defined.

In a study of a rat model of contrast-associated AKI induced by dehydration and iohexol, 38 circRNA transcripts were differentially expressed, with 16 circRNAs upregulated and 22 circRNAs downregulated, compared with the control rats treated with saline. Several candidate miRNAs were linked with the differentially expressed circRNAs in this study, and pathway analyses showed that these circRNAs were typically associated with DNA replication and steroid hormone biosynthesis as well as chemical carcinogenesis [89]. However, the predicted gene products of these ncRNA pathways were not assayed in the rat models, nor were these findings validated against human subjects presenting with contrast-associated AKI.

## 6. CircRNA Expression in AKI Due to Urinary Tract Obstruction

Obstruction occurring anywhere along the urinary tract can lead to an acute reduction in urine output and glomerular filtration rate. Bladder outlet obstruction or bilateral ureteric obstruction blocks off the urinary outflow of both kidneys, leading to complete renal shutdown, but unilateral ureteric obstruction can also lead to a reduction in kidney function, even in the contralateral unobstructed kidney, through a number of inflammatory and vasoactive pathways [90]. Typically, a triphasic response of renal blood flow and ureteral pressure is observed in ipsilateral ureteric obstruction. Within the first 1.5 h, a strong prostaglandin response is evoked due to rising ureteral pressures, leading to an increase in renal blood flow. In the second phase, which lasts about 3 to 4 h, renal blood flow decreases due to the effects of thromboxane A2, and the renin–angiotensin–aldosterone system is also progressively activated, resulting in renal vasoconstriction and a reduction in renal blood flow, though ureteral pressures still remain elevated. In the third phase, both the renal blood flow and the ureteral pressure drops due to the loss of glomerular filtration rate [91,92]. The first two phases also affect the contralateral kidney since vasoactive pathways are invoked. Furthermore, in addition to changes in blood flow, as tubules dilate and the interstitium expands in obstructed kidneys, there is a time-dependent increase in leukocyte infiltration, often promoted by the MAPK pathway, resulting in epithelial cell death and the activation of myofibroblasts [93]. Significant and irreversible scarring tends to occur from prolonged urinary tract obstruction, with the activation of TGF-β1-related pathways, resulting in an epithelial-to-mesenchymal transition (EMT) and the development of renal fibrosis [90].

In animals, the unilateral ureteric obstruction (UUO) model was first proposed to study the effects of obstructive nephropathy. This typically involves a simple surgery in the target animals whereby the ureter of one kidney is ligated, with subsequent development of hydronephrosis, tubular cell death, inflammation, and the development of renal fibrosis. The unobstructed, contralateral kidney could be studied as a control after the animals involved were sacrificed [94]. Although this model is effective at assessing the effects of ureteric obstruction on renal fibrosis, with potent induction of TGF-β1-related fibrotic pathways, ascertaining the effects on the microvasculature is more problematic as the contralateral kidney is also affected by changes in neurohormonal pathways [95]. Thus, most studies of ncRNAs including circRNAs, using the UUO model, have focused on the subsequent development of renal fibrosis and accrual of chronic damage, rather than on the shorter-term effects and outcomes *per se*.

RNA-sequencing showed 63 upregulated and 64 downregulated circRNAs in human patients with ureteropelvic junction obstruction. For illustration, the expression of hsa_circ_0045861 was significantly upregulated, and this was accompanied with a resultant decrease in miR-181d-5p. The increase in hsa_circ_0045861 was simulated in vitro by the treatment of HK-2 cell lines with TGF-β1, suggesting that the circRNA identified is associated with TGF-β1-mediated renal fibrosis [56]. These findings mirror those in other mammalian models, with 605 circRNAs upregulated and 745 circRNAs downregulated in a murine UUO model. Gene Ontology analysis showed that many of these circRNAs were involved in pathways relating to cell apoptosis and p53 signalling [96]. In a smaller study, mmu_circ_30032 expression was upregulated in the murine UUO model, promoted by the activation of the p38-MAPK pathway, resulting in the deposition of type I collagen, type III collagen, and fibronectin via miR-96-5p [57]. Circ-ACTR2 (hsa_circ_0008529) was a sponge for miR-561 and promoted the activation of NLRP3, pyroptosis, and inflammation in macrophages, thereby inducing EMT and renal fibrosis through paracrine effects [58]. Conversely, mmu_circ_37492 was increased and possibly protective against renal fibrosis in a murine UUO model by sponging miR-7682-3p, therefore upregulating its downstream target Fgb. The human homologue to mmu_circ_37492, designated as hsa_circ_0012138, may also target renal fibrosis in a similar manner via the sponging of miR-651-5p [59].

## 7. Potential Clinical Applications of circRNAs in AKI

CircRNAs are more stable than other ncRNAs such as miRNAs, making them attractive as biomarkers for diagnosis and prognostication. The wide array of differentially expressed circRNAs in both human subjects as well as murine models will have to be validated in prospective trials to ascertain their relationship to important clinical outcomes. As an illustration, as aforementioned, circulating ciRs-126 levels at the start of the first dialysis session showed a logarithmic relationship with short-term mortality in dialysis-requiring AKI patients in the ICU, although this was only a single-centre study with a small study cohort with severe acute kidney injury [29]. Given that the pathogenesis differs between causes and syndromes of AKI, these biomarkers will have to be validated in specific AKI syndromes. While most studies of circRNAs in AKI have focused on blood and tissue expression of circRNAs, it is also prudent to note that these circRNAs may not be specific for kidney disease *per se*, especially if they are assayed from the blood. As mentioned previously, like other ncRNAs, circRNAs can be identified in exosomes released into a variety of body fluids. Experimentally, circ-FANCA could be identified from exosomes released from HK-2 cells treated with LPS [31]. CircRNAs are retrievable from urine samples, and these may prove to be better sources of molecular information regarding kidney diseases including AKI [97].

Compelling in vitro data using human kidney cell lines showed that silencing certain upregulated circRNAs or transfection with downregulated circRNAs could both have salutary effects on tissue injury, particularly in septic AKI or IRI. Interestingly, this could be effectuated using non-specific agents such as losartan in the rat model of IRI [54], or by the design of agents specifically targeted at certain circRNAs, as illustrated in Figure 3. Targeted methods that are being tested include the use of interfering RNA, the delivery of circRNA overexpression via lentivirus- or adenovirus-based vectors, the delivery of circRNA via nanoparticles such as liposomes, the creation of synthetic circRNAs, and the administration of exosomes containing circRNAs, among other methods [98]. Encouragingly, intravenous administration of mmu_circ_0001295 derived from hypoxia pre-treated adipose-derived mesenchymal stem cell (ADSC) exosomes directly into CLP mice led to the amelioration of tissue inflammation and injury and the attenuation of kidney dysfunction [99]. Synthetic circRNAs have also been constructed for the treatment of other diseases, although they remain at a very early stage of development [100,101,102]. The safety profile of circRNA treatment needs to be carefully examined in future clinical trials since the expression of many circRNAs are not limited to kidney tissue, which may cause untoward off-target effects. Additionally, interfering RNAs used for the treatment of other diseases may sometimes silence off-target genes as well as via an miRNA-like effect [103]. Synthetic circRNAs may also be potentially immunogenic when administered to humans [102]. Evidently, the development of therapeutics targeted at circRNAs will be technically challenging.

## 8. Conclusions and Future Perspectives

AKI is a common clinical condition and important cause of morbidity and mortality. There is an urgent need for better diagnostic, prognostic, and therapeutic tools in different syndromes of AKI. In addition to conventional proteomic and genomic approaches, the characterisation of the epigenetic landscape in AKI has opened the possibility of harnessing ncRNAs for the evaluation and treatment of various diseases. Compared to miRNAs, circRNAs are more stable, and are readily detectable by modern sequencing tools, suggesting that they would be potential candidates for liquid biopsy and other non-invasive diagnostic strategies. Various ncRNAs including circRNAs are differentially expressed in AKI compared to the healthy controls. The characterisation of ncRNA-associated competing endogenous RNA (ceRNA) networks in various models of AKI would help to identify ncRNAs that can be targeted therapeutically to attenuate disease processes. The identification of novel delivery systems for circRNA-based therapeutics would be crucial to translate circRNAs from the bench to the bedside. With a greater understanding of their roles in the pathogenesis and treatment of AKI, circRNAs could eventually become important tools for the diagnosis and treatment of AKI in the years to come.

## Figures and Tables

**Figure 1 ijms-23-08509-f001:**
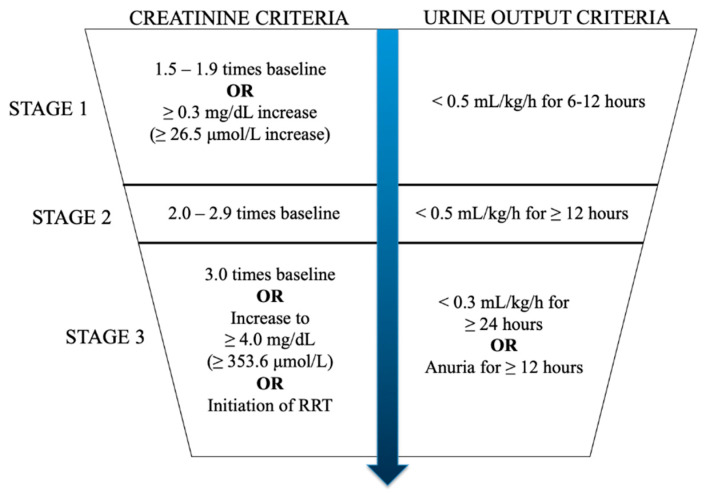
The KDIGO criteria for the diagnosis of acute kidney injury (AKI). *RRT: renal replacement therapy*.

**Figure 2 ijms-23-08509-f002:**
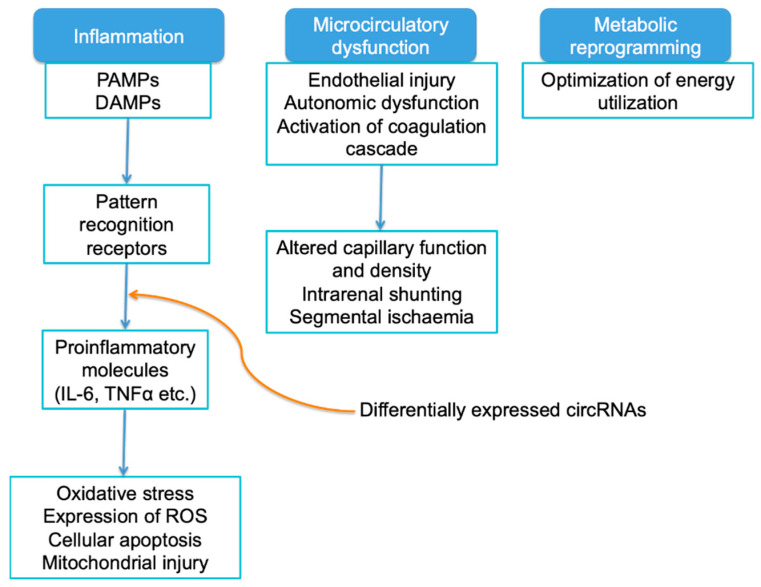
A schematic diagram of the pathophysiology of septic acute kidney injury (AKI). DAMP: damage-associated molecular patterns; PAMP: pathogen-associated molecular pattern; ROS: reactive oxygen species.

**Figure 3 ijms-23-08509-f003:**
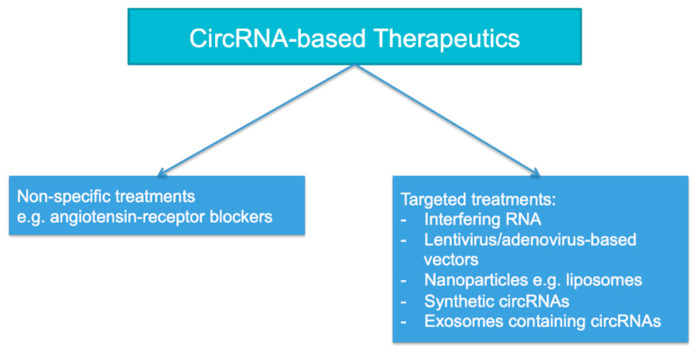
The therapeutic options for acute kidney injury (AKI) based on circRNA.

**Table 1 ijms-23-08509-t001:** Differentially expressed circRNAs in different syndromes of acute kidney injury (AKI).

AKI Syndrome	CircRNA	Dysregulation
Undifferentiated	hsa_circ_0003266/ciRs-126	Upregulated [29]
Sepsis	hsa_circ_0040994/circ-FANCA	Upregulated [31]
	hsa_circ_0002131/circ-BNIP3L	Upregulated [32]
	Circ-HIPK3	Upregulated [33,34]
	hsa_circ_0114427	Upregulated [35]
	hsa_circ_0114428	Upregulated [36]
	Circ-TLK1	Upregulated [37]
	Circ-RASGEF1B	Upregulated [38]
	hsa_circ_0068,888	Downregulated [39]
	hsa_circ_0091702	Downregulated [40]
	Circ-PRKC1	Downregulated [41]
	Circ-Ttc3	Downregulated [42]
	hsa_circ_0007847/circ-MTO1	Downregulated [43]
	Circ-VMA21	Downregulated [44]
Ischaemia-reperfusion injury	hsa_circ_0023404	Upregulated [45]
	Circ-AKT3	Upregulated [46]
	hsa_circ_0018148/circ-ITGB1	Upregulated [47]
	hsa_circ_001839	Upregulated [48]
	mmu_circ_0000943	Upregulated [49]
	Circ-SNRK	Upregulated [50]
	mmu_circ_0000823/circ-SLC8A1	Upregulated [51]
	mmu_circ_0014064/circ-APOE	Upregulated [52]
	Circ-YAP1	Downregulated [53]
	Circ-DNMT3A	Downregulated [54]
	Circ-PLEKHA7	Downregulated [20]
	Circ-ME1	Downregulated [52]
Cisplatin nephrotoxicity	hsa_circ_0114427	Upregulated [55]
Urinary tract obstruction	hsa_circ_0045861	Upregulated [56]
	mmu_circ_30032	Upregulated [57]
	hsa_circ_0008529/circ-ACTR2	Upregulated [58]
	mmu_circ_37492	Upregulated [59]
	hsa_circ_0012138	Upregulated [59]

## Data Availability

Not applicable.

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
