# Peer review of "Circular RNAs in Acute Kidney Injury: Roles in Pathophysiology and Implications for Clinical Management"

_ijms, 2022, doi:10.3390/ijms23158509_

Round 1

Reviewer 1 Report

Circular RNAs (circRNAs) are endogenous RNA that are produced by selective RNA splicing and are involved in the progression of various diseases. Studies have shown that various kidney diseases, including renal cell carcinoma, acute kidney injury, and chronic kidney disease, are linked to circRNAs. In this review article, So et al, are focusing on circRNAs in Acute Kidney Injury, their role in pathophysiology, and implications for clinical management.

The subject is relevant, unknown to many, but with significant future perspectives. 

The review is well organized and comprehensively described with adequate and not misleading references.

I have only the following minor suggestions for the authors.

i)                 shortening the introductory paragraph (some concepts concerning AKI are repetitive).

ii)                change the title of the second paragraph; from CircRNAs in Healthy and Diseased Kidneys to CircRNAs in Healthy and Kidney-Related Diseases

Author Response

  1. The introductory paragraph has been edited to reduce redundancy.
  2. The title has been changed to CircRNAs in Health and Kidney Diseases.

Reviewer 2 Report

The aim of this manuscript is to review the relationship and potential mechanism of circular RNAs in the pathological process of AKI. The author reviewed the pathology of AKI and the molecular biology of circRNAs. They also reviewed expression of circRNA in different etiology of AKI including sepsis, ischemia-reperfusion injury and drug-induced AKI.

1. AKI can be classified as Prerenal, Intrarenal and Postrenal. It would be better for the author to reorganize this review article based on this classification which would be easy for the reader.

2. Although Postrenal AKI (Urinary tract obstruction) only account for about 10% of AKI, several animal models has been used to study this etiology, which should be reviewed and discussed in this manuscript. 

3. A schematic with the potential mechanism of CircRNA in AKI would be helpful for the readers.

4. Further detail for the association between CircRNA and Dialysis-required AKI in the ICU should be reviewed and provided in Section 6. 

Author Response

  1. I reference the classical categorization into "pre-renal, renal and post-renal" in the updated manuscript. Since a) the latest KDIGO guidelines for acute kidney injury recommend identifying specific aetiologies, and b) the pathophysiology behind different aetiologies is complex and there are often "pre-renal", "renal" and "post-renal" components, and c) the literature surrounding circRNAs in AKI are built around specific animal models or specific populations, who are categorized according to the aetiology, the review was organized around these aetiologies / models.
  2. The updated manuscript includes a new section specifically devoted to obstructive AKI, though recognizing that most of these were built on the unilateral ureteric obstruction (UUO) model which aimed to use ureteric obstruction to simulate renal fibrosis in CKD, rather than to model obstructive AKI per se.
  3. The exact mechanisms of many differentially expressed circRNAs were not well-described in the literature, especially in network analysis where high throughput sequencing was used to quickly screen for differentially expressed circRNAs. However, a schematic explaining the pathophyisology of AKI in the most well-described scenario of sepsis, specifically highlighting that most differential expression of studied circRNAs is focused on their effects on expression of inflammatory mediators.
  4. Additional detail regarding this specific study has been furnished; this is the same study referenced and explained in the last paragraph of section 2.

Reviewer 3 Report

In the manuscript, IJMS-1823909 by So et al, authors have provided a literature review on the roles of Circular RNAs in Acute Kidney injury. Authors have discussed the evidence for the function of CircRNAs in different human and murine models of Acute kidney injury. Potential application of CirRNAs for the diagnosis, prognostication and treatment of Acute kidney injury have been discussed. Along with common occurrence, Acute kidney injury (AKI) is associated with significant morbidity, mortality. Understanding role of CirRNAs in the pathogenesis and treatment of Acute kidney injury would add to the therapeutic devilment. Authors efforts are well reflected in the manuscript. In general review is well written and the subject covered in important and interesting.

There are some suggestions to improve this review-

1.     Line 7.  “Acute Kidney injury is common” Please rephrase. For instance, …. “is common disease condition/syndrome”

2.     Please consider extending the legend of figure 1 a bit more.

3.     In the introduction section "Circular RNAs" part is very concise. Authors can add a little more literature about CirRNAs in general animal development disease conditions.

4.     In addition to the above comment. The last paragraph of the introduction is important but weaker in terms of providing proper logics/rationale for this review.

5.     Manuscript contains circRNAs and CircRNAs. Do authors imply different things with cirRNAs and CirRNAs? If yes, pls mention, if not, please consider using a consistent abbreviation

6.     This review is very descriptive and important. I would expect this review to be supplemented with some schematic diagrams for a quick and easy understanding for the reader. Schematics can be drafted for at least for section 3, 4 and 6.

7.     It would be very informative to have some future perspectives in this subject. Authors can speculate some future directions to stretch over the scope of further research.

Author Response

  1. The abstract has been updated as "Acute kidney injury is a common condition"
  2. The legend of figure 1 makes reference to the only abbreviation inside the text of the diagram; the criteria themselves are self-explanatory otherwise.
  3. The text has been reorganized such that some segments from part 2 (circRNAs in Health and Kidney Disease) is now moved to the introduction in part 1 instead.
  4. The last paragraph of the introduction has been improved to explain the rationale for reviewing the role of circRNAs.
  5. "CircRNAs" was capitalized when the term was used at the start of sentence, and rendered as "circRNAs' when it occurred in the middle of a sentence.
  6. Schematic diagrams have been added for the section on "circRNAs in septic AKI", and for "circRNA-based therapeutics".
  7. Some additional future perspectives have been added to the last paragraph as suggested.